# Iron Metabolism and Ferroptosis in Physiological and Pathological Pregnancy

**DOI:** 10.3390/ijms23169395

**Published:** 2022-08-20

**Authors:** Yijun Zhang, Yun Lu, Liping Jin

**Affiliations:** Shanghai Key Laboratory of Maternal-Fetal Medicine, Department of Reproductive Immunology, Shanghai First Maternity and Infant Hospital, School of Medicine, Tongji University, Shanghai 200092, China

**Keywords:** iron metabolism, ferroptosis and pregnancy

## Abstract

Iron is a vital element in nearly every living organism. During pregnancy, optimal iron concentration is essential for both maternal health and fetal development. As the barrier between the mother and fetus, placenta plays a pivotal role in mediating and regulating iron transport. Imbalances in iron metabolism correlate with severe adverse pregnancy outcomes. Like most other nutrients, iron exhibits a U-shaped risk curve. Apart from iron deficiency, iron overload is also dangerous since labile iron can generate reactive oxygen species, which leads to oxidative stress and activates ferroptosis. In this review, we summarized the molecular mechanism and regulation signals of placental iron trafficking under physiological conditions. In addition, we revealed the role of iron metabolism and ferroptosis in the view of preeclampsia and gestational diabetes mellitus, which may bring new insight to the pathogenesis and treatment of pregnancy-related diseases.

## 1. Introduction

Iron is essential for a variety of biological processes including oxygen transport, ATP production, DNA biosynthesis, and cell proliferation. During pregnancy, iron requirements increase dramatically as the maternal blood volume expands and the fetus develops, making iron deficiency the most common nutrient deficit worldwide [1]. Iron deficiency is correlated with several adverse pregnancy outcomes, such as increased maternal illnesses, prematurity, and intrauterine growth restriction. Given that maintaining sufficient maternal iron level is of great importance, iron supplementation is almost universally recommended in pregnant women [2].

As a “gatekeeper” of iron between the mother and fetus, placenta plays a pivotal role in iron transport during pregnancy. In species with hemochorial placentas, such as human and rodents, iron in maternal circulation must be transferred to the fetus through placental syncytiotrophoblasts (STB) [3]. Although the identification of several iron transporters and iron-regulating hormones has greatly improved our understanding of cellular iron transport and regulation, the precise mechanism regarding iron trafficking in STB and fetal endothelium still remains obscure. Furthermore, placental iron trafficking can be regulated by maternal, fetal, and placental signals. Rising evidence demonstrating that the placenta prioritized its own metabolic needs in the face of severe iron deficiency has challenged the traditional dogma of exclusive fetal-demand-driven iron transfer process [4,5], which will be discussed in detail later.

Like most other nutrients, iron exhibits a U-shaped risk curve [6]. Apart from iron deficiency, iron overload is also dangerous since labile iron is redox-active and toxic. It can generate reactive oxygen species (ROS), which results in oxidative stress and activation of programmed cell death pathways, such as ferroptosis and iron-related autophagy [7]. Ferroptosis is characterized by iron accumulation and lipid-peroxidation-mediated cell membrane damage. It has been related to diseases including cancer, neurodegeneration, and myocardial infarction, yet its role in pregnancy-related disorders has not been fully elucidated. In this review, we mainly focus on the molecular mechanism and regulation of placental iron trafficking, as well as elucidating the role of iron metabolism and ferroptosis in the view of preeclampsia and gestational diabetes mellitus based on current knowledge.

A search of the PubMed database was performed to identify relevant literatures written in English focusing on iron metabolism and pregnancy. The search term “iron metabolism” was combined with a range of terms related to the focus of the review, such as pregnancy, placenta, placental trafficking, homeostasis, systemic regulation, ferroptosis, preeclampsia, and gestational diabetes mellitus. The search included studies from 1975 to 2022. After title and abstract screening for study type, case reports, case series, and animal studies with non-rodents were excluded. The reference list was further checked by full-text reading, and 141 articles were included in the final review.

## 2. Brief Understanding of Physiological Iron Metabolism

### 2.1. Basic Steps in Systemic Iron Transport

Successful transport of iron requires the coordination of various related proteins, including divalent metal transporter 1 (DMT1) [8], which supports intestinal iron absorption and iron transport through acidified endosomes [9]; ferroportin (FPN), the sole membrane iron exporter [10,11]; transferrin (Tf), an iron-binding serum protein [12]; transferrin receptor 1 (TfR1), which induces cellular uptake of iron from Tf by endocytosis [13]; ferritin, an intracellular iron storage protein [14]; and hepcidin, a circulating peptide hormone regulating iron homeostasis by mediating the degradation of FPN [15,16].

Under physiologic circumstances, all iron entering the body is from the diet. It is mainly absorbed by the villus of mature enterocytes at the proximal segment of the small intestine, where the low pH helps to keep it in a soluble form, thus making it available for absorption. Since most dietary iron is in the ferric (Fe^3+^) form, it needs to be reduced to the ferrous (Fe^2+^) form by reductase such as duodenal cytochrome B [17] before being imported by DMT1 at the apical membrane of enterocytes. If iron in the enterocytes is temporarily not required, it will combine with ferritin and become lost from the body once the enterocytes reach the end of their life span. Otherwise, the iron will be rapidly exported to plasma by FPN on the basolateral membrane of enterocytes after being oxidized by iron oxidase hephaestin [18]. In the plasma, iron is bound to Tf, which can bind 2 atoms of iron at most and deliver them to the sites of utilization. Normally, the plasma transferrin saturation is around 30%, providing a powerful buffering ability for potential toxic iron overload [19]. Apart from dietary iron absorbed by enterocytes, other resources of plasma iron include macrophages and hepatocytes. The former release heme-derived iron from senescent red blood cells and the latter is the largest storage site of iron in the body.

Plasma diferric Tf transports iron into the cells by binding with TfR1 expressed on the membrane of almost every cell. The Tf–TfR1 complex is internalized by receptor-mediated endocytosis (Figure 1). Within the highly acidified endosome, the transformation of Tf and reduction of Tf-bound ferric iron takes place with the help of ferrireductase STEAP3 [20], resulting in the release of iron from Tf [21]. The iron crosses the endosomal membrane into the cytoplasm via DMT1. Since free iron in solution is quite dangerous, it is reasonable to hypothesize that iron-binding proteins or chaperones exist in the cytoplasm to move iron within the cell. The newly discovered intracellular iron chaperone was poly (rC)-binding proteins, which showed the ability to transfer cytoplasmic iron to ferritin and several enzymes [22]. The fate of intracellular iron depends on the demands of cells. If iron is currently requested in the metabolic reactions, it may be transported directly to the sites of need such as the mitochondria. Otherwise, it would be sequestered in a non-toxic form with ferritin, providing an iron reservoir for later use. Ferritin, a large spherical protein consisting of 24 subunits, can hold up to 4500 molecules of iron, which makes it indispensable for maintaining cellular iron homeostasis [23]. Ferritin can also be secreted into the plasma. The plasma ferritin concentration strongly corelates with intracellular iron storage level, making it a good indicator of body iron stores. Upon mobilization, iron can efficiently detach from ferritin and be exported by FPN, which also serves as a “safety valve” in case of iron overload taking place intracellularly. 

### 2.2. Regulation of Iron Homeostasis

Iron is pivotal for many biological processes but is also toxic in excess. Thus, body iron balance needs to be sophisticatedly regulated. On the systemic level, the main character modulating iron homeostasis is a hepatic hormone hepcidin. It was first identified as an antimicrobial property [24] and then was discovered to control the release of iron from cells. Hepcidin binds with ferroportin, the only known membrane iron exporter, and leads to the internalization and degradation of it [15], resulting in decreased cellular iron egress and reduced plasma free iron concentration. A recent study on hepcidin structure revealed that only ferroportin molecules loaded with iron are targeted for degradation [25], demonstrating a regulatory strategy with high selectiveness. Three main functions of hepcidin include maintaining stable iron stores, providing enough substrates for erythropoiesis, and limiting iron availability to microorganisms. Since hepcidin is also under the control of the classical endocrine feedback system [26], the factors regulating its production lie accordingly with its functions. Hepcidin production is decreased by iron deficiency and erythropoietin in support of erythropoiesis and other physiological needs, while its production is increased by iron overload and inflammation in order to protect cells from toxic effects. Dysregulation of hepcidin is correlated with several pathological conditions. Hepcidin deficiency or resistance contributes to iron overloading disease such as hereditary hemochromatosis, whereas excessive hepcidin is associated with iron-restriction anemia in chronic inflammatory diseases.

At the cellular level, iron homeostasis is mainly maintained by iron regulatory proteins (IRPs). IRPs can regulate the transcription of iron-related proteins by binding to the iron-responsive element (IRE) region in the untranslated regions (UTR) of their mRNAs (Figure 1). When cellular iron concentration is low, IRPs would bind to the 5′ UTR of ferritin mRNA and the 3′ UTR of TfR1 mRNA, which blocks the translation of the former and protects the latter from degradation [27]. Taken together, it keeps intracellular ferritin at a low level when iron storage is not needed, while ensuring more expression of TfRs on the cell surface, maximizing the iron input. Conversely, IRPs would not bind to the IRE at a high cellular iron level, thus allowing the translation of ferritin and exposing TfR1 mRNA for degradation, which prevents iron import and supports iron storage. The main purpose of this IRP-IRE regulating system is to guarantee that cells are neither iron-deficient nor iron-replete. It is the most studied mechanism but not the only one. Other factors such as hypoxia, hormones, and cytokines were also proved to exert regulatory effects on iron related genes [28].

## 3. Adaptions of Iron Metabolism in Physiological Pregnancy

### 3.1. Requirements of Iron during Pregnancy

Iron requirement increases dramatically during pregnancy to support maternal blood volume expansion and fetal development. Approximately 1 g of additional elemental iron is needed every day during pregnancy [29]. Apart from supporting maternal hemoglobin synthesis, adequate iron concentration in utero is also essential for fetal development and helps build iron stores in early infancy [30]. Since iron deficiency has been associated with adverse maternal-fetal outcomes such as increased maternal morbidity, preterm birth, intrauterine growth restriction, and neurocognitive impairment in offspring, iron supplement during pregnancy is universally recommended [31,32].

### 3.2. Effects of Gestational Iron Deficiency on the Fetus

Moderate to severe gestational iron deficiency could lead to both immediate and long-term consequences to the offspring. Given the important role played by iron in neurodevelopment, brain structure alterations and cognitive impairments are the most common short-term risks [33]. Evidence from animal models and human observational studies have demonstrated that the hippocampus and the process of myelination are particularly vulnerable to iron deficiency [34]. The imbalance between high energy demand in these rapidly growing structure and impaired cellular energy metabolism in the lack of iron could be the possible explanation. Moreover, gestational iron deficiency may also be associated with long-term mental health disorders such as depression, anxiety, autism, and schizophrenia in offspring [35]. This could be accounted for by decreased monoamine production with impaired emotion and memory process due to iron deficiency [36]. Iron supplementation during the gestational and postnatal periods may be beneficial to neurodevelopment [37].

### 3.3. Iron Transport by Placenta

In both humans and rodents, iron in maternal circulation is transferred to the fetus directly through placental syncytiotrophoblasts (STB). The major source of iron is transferrin-bound iron (TBI), which is imported by TfR1 on the apical side of STB and endocytosed into the cell [38,39]. The acidified endosomes release iron into the cytosol by DMT1 [40]. The iron can either be used or stored within the placenta or be exported through FPN on the basal layer of STB, which enters the fetal circulation after crossing the fetal epithelium. This is the most well-studied process of iron trafficking through the placenta [41,42] and is in parallel with that observed in other cells (Table 1).

However, certain questions still remain regarding the precise mechanism of placental iron transport. For example, some researchers argued that DMT1 is not needed for maternal–fetal iron transfer since Dmt1-null mice exhibited similar body iron content comparing with wildtype mice, indicating that placental iron transfer was still efficient [45]. One possible explanation is that alternative endosomal iron transporters exist except for DMT1. ZIP14 (SLC39A14) and ZIP8 (SLC39A8) both belong to the SLC39A metal-ion transport family and have been identified as iron transporters recently. They were proven to mediate the uptake of non-transferrin-bound iron (NBTI) on the plasma membrane [61] and the cellular assimilation of iron in endosomes [62]. ZIP14 is expressed in mouse placenta [63] while its expression profile in human placenta is still lacking [64]. Meanwhile, ZIP8 was found to be highly expressed in human placenta cells. Suppression of endogenous ZIP8 expression in BeWo cells, a placental cell line, reduced iron uptake by 40% [65]. Global knockout of Zip8 in mice is embryonically lethal, suggesting an essential role played by ZIP8 in early development [66]. Taken together, ZIP14 and ZIP8 may participate the placental iron transport process, but further validation is still needed.

Another question is what happens after iron exiting from STB. In human placenta, a single layer of STB separates maternal blood from the fetal capillary endothelium and facilitates the transport of all nutrients and waste [3]. It is reasonable to speculate that a small fraction of iron may enter fetal circulation through the NTBI pathway while the majority of iron still needs to be bounded by Tf. Ferrous iron exported by FPN must be oxidized to ferric state before binding with Tf. This process is mediated by multicopper ferroxidases such as hephaestin, ceruloplasmin, and zyklopen (Zp) [67], among which Zp was the only one found to be abundantly expressed in human and mouse placentas [53,68]. The distinct expression pattern and confirmed oxidase activity of Zp suggested it could be essential for placenta iron transport. Nevertheless, a recent study disproved this idea by showing that the amount of iron transferred to the fetus remained unchanged in Zp knockout mice [54]. It was also demonstrated that Zp is predominately localized in maternal decidua rather than the nutrient-transporting STB [54]. Future studies regarding how iron is trafficked through the fetal endothelium should be of interest. 

Lastly, apart from TBI, other forms of circulating iron such as NTBI, ferritin, and heme may also take a part in placental iron trafficking. The most widely studied NTBI transporters are ZIP14 [69] and ZIP8, whose expressions have been identified in placenta tissues [63,65]. However, research concerning NTBI transport within the placenta are extremely limited given that NTBI is a redox-active and potentially toxic form of iron, one that is quickly cleared from the plasma by the liver [70]. Investigation on this topic would be of great importance to patients in iron overload conditions since NTBI may be untaken by fetal neurons, resulting in generation of ROS and neonatal encephalopathy [71,72]. Another source of iron could be ferritin in maternal circulation. Early in the 1970s, maternal–fetal ferritin transport was elucidated by intravenous injection of isotope-labeled ferritin in pregnant rabbits and guinea pigs. Electron microscopy showed the accumulation of ferritin particles in placental basement membrane, fetal capillaries [73], and endocytic vesicles [74], suggesting the endocytosis of ferritin. The existence of ferritin has also been confirmed in human STB microvillus plasma membrane [75]. Recently, TfR1 [76] and a novel receptor called SCARA5 [77] were identified to mediate cellular ferritin uptake, which unrevealed an alternative mechanism of iron delivery. Furthermore, heme iron, which is either released to plasma by erythrocyte catabolism or directly exported by cells with heme exporters, may also be utilized in placental iron trafficking. Numerous heme transporters or metabolic enzymes, such as feline leukemia virus subgroup C receptor 1 (FLVCR1), breast cancer resistance protein (BCRP) [78], LDL-receptor-related protein 1 (LRP1) [79], and heme oxygenase (HO)-1 [59], were found to be richly expressed in placentas. Data are lacking on the functions of these proteins in placenta, and the way in which heme metabolism participates in maternal–fetal iron transport requires further elucidation.

### 3.4. Regulation of Placental Iron Trafficking

Since iron deficiency during pregnancy can pose a problem for both maternal health and fetal development, understanding the regulation of placental iron trafficking would be beneficial to the prevention and treatment of fetal iron deficiency or anemia. As a “gatekeeper” of iron transfer, the placenta balances the competition of iron between mother and fetus over a limited amount of supply, which is mainly achieved by regulating the expressions of iron transporters on STB.

The upregulation of placental TfR1 in response to maternal iron deficiency in human and animal models has been observed in multiple studies [5,80,81,82]. It was interpreted as a compensatory mechanism employed by the placenta to ensure adequate iron delivery to the fetus. One possible explanation is that the binding of IRPs to the 3′ UTR of TfR1 mRNA can protect it from degradation during iron deficiency [83]. However, Sangkhae et al. discovered that the expression of TfR1 was only slightly affected when IRP1 was significantly upregulated in iron-deficient mice placentas. Meanwhile, TfR1 showed no obvious difference in iron-deficient IRP1-KO and WT mice [5]. This could be attributed to the low expression of ribonuclease regnase 1, which degrades the TFR1 transcript under iron-sufficient conditions [84], but further investigation is certainly needed.

Maternal iron availability in the circulation is essential for iron transfer. Since the demand for iron increases tremendously along with gestational age [85], additional iron must be absorbed from the diet or be mobilized from the iron storage [86,87]. This process is mediated by hepcidin that interacts with cellular iron exporter FPN [88]. A profoundly decreased level of hepcidin was found during the second and third trimesters of pregnancy in healthy women [89] and rodents [90], thus increasing the iron export from cells such as duodenal enterocytes, recycling macrophages, and hepatocytes, which maximizes the iron available for placenta. Although the exact mechanism of hepcidin suppression remains unknown, researchers have proven that maternal hepcidin determines iron endowment in embryos [4]. In Hamp knockout pregnant mice, the non-heme iron level was found to be significantly increased in both placentas and embryo livers. Injecting a high dose of hepcidin agonist caused anemia, tissue iron deficiency, and increased mortality in embryos [4]. Apart from maternal hepcidin, fetal liver can also produce hepcidin. Given that placenta FPN is located on the fetal facing-side of STB, it is reasonable to presume that it can be regulated by fetal hepcidin to maintain iron homeostasis. Surprisingly, there was no significant difference in iron status or placental FPN expression between Hamp ablated and wildtype embryos, suggesting that fetal hepcidin could not regulate placenta iron export [4,5]. However, it can be argued that fetal hepcidin activity was just too low to exert the expected effect since the suppression of placental FPN was seen in transgenic fetal hepcidin overexpression models [91,92]. The major role of fetal hepcidin may lie in the autocrine/paracrine effect on FPN in hepatocytes, resulting in preserved fetal liver iron store and promoted erythropoiesis [92]. 

Over the past 20 years, evaluation of the relationship between maternal iron state and fetal iron endowment at birth has increased remarkably. Moreover, the most frequently asked question is: who is prioritized in iron deficiency? The observation that iron-replete neonates (measured by cord blood ferritin) were born by iron-deficient women [93,94] indicates the prioritization of fetal iron needs. It is not surprising that the placenta and fetus were described as “a parasite upon mother” since they acquire adequate iron despite the limited maternal iron status. Nevertheless, a recent study demonstrated that the placenta becomes “selfish to the fetus” when iron could not fulfill its own metabolic needs. In iron-deficient mice, placental FPN expression significantly decreased during the whole gestational period, which compromised iron delivery to the fetus [5,95]. It could be accounted to the increased binding of IRP1 to the IRE in 5′-UTR of Fpn1A transcript, resulting in the suppression of Fpn1A translation. This phenomenon is unique to placental STB since the non-IRE Fpn1B transcript is predominantly expressed in other cells such as enterocytes and macrophages [96]. Maintaining the iron hemostasis is essential to support trophoblast function, and severe placental iron deficiency could impair cellular oxidative phosphorylation [5,97]. All conclusions above were drawn with animal data, and further validation on the human placenta is needed [98]. 

In conclusion, placental iron trafficking can be regulated by maternal, fetal, and placental signals. Under most circumstances, maternal iron status and placental iron transporters were regulated in the favor of fetal requirements, while the placenta prioritized its own metabolic needs in the face of severe iron deficiency.

## 4. Ferroptosis: An Iron-Related Programmed Cell Death Pathway

Ferroptosis is a new type of programmed cell death that occurs with iron dependence. In 2012, Dixon et al. firstly proposed the concept of ferroptosis [99], which is characterized by iron accumulation and lipid-peroxidation-mediated cell membrane damage. Morphologically, ferroptosis mainly manifests abnormalities in mitochondria such as the loss of structural integrity and increased membrane density, which obviously differs from apoptosis, pyroptosis, necroptosis, and autophagy [99,100]. On the molecular level, glutathione peroxidase 4 (GPX4) is a major anti-oxidative enzyme that prevents lipid peroxidation caused by ROS. The protective role of GPX4 requires the participation of glutathione (GSH) and Cys2/glutamate antiporter (SLC7A11) [101]. Depletion of intracellular glutathione (GSH) and inactivation of glutathione peroxidase 4 (GPX4) inhibits the antioxidant defense of the cell. The ultimate oxidation of lipids in an iron-dependent manner results in massive production of ROS, thus promoting ferroptosis [102]. 

Intracellular iron accumulation is closely related to ferroptosis. Iron can directly generate an excessive amount of ROS through Fanton action, which causes oxidative damage [99]. In addition, iron may also increase the activity of lipoxygenases (LOXs) that are responsible for lipid peroxidation and the initiation of ferroptosis [103,104]. These processes are strictly correlated with cellular labile iron because inert iron stored with ferritin could not contribute to lipid peroxidation. Hence, ferritin concentration becomes a key target to regulate the sensitivity of ferroptosis. It has been demonstrated that ferritinophagy, a ferritin-targeted autophagy, which leads to the lysosomal degradation of ferritin and the release of labile iron, caused increased sensitivity to ferroptosis [105]. Overexpression of the ferritinophagy selective receptor nuclear receptor coactivator 4 (NCOA4) promoted ferroptosis by increasing ferritin degradation [106]. Other systemic or local iron regulation signals may also affect ferroptosis.

The most common lipids for initiating ferroptosis belong to the polyunsaturated fatty acid (PUFA) family [7]. Therefore, lipid synthesis of PUFA increases the cellular sensitivity of ferroptosis while fatty acid beta-oxidation, which consumes a large number of lipids, has the opposite effect. As a major characterization of ferroptosis, lipid peroxidation refers to a process by which oxidants attack the double bond of membrane lipids. However, where exactly lipid peroxidation occurs during ferroptosis remains obscure. Many cellular components, including the plasma membrane [106], mitochondria [102], endoplasmic reticulum, lysosomes [107], and nucleus membrane [108], could be the potential targets of lipid peroxidation. Understanding how lipid peroxidation induces cell death is still a huge challenge, and the newly discovered inducers, inhibitors, and pathways related to ferroptosis have been reviewed elsewhere [109].

## 5. Role of Iron Metabolism and Ferroptosis in Pathological Pregnancy

Dysregulation of iron metabolism and ferroptosis has been related to multiple diseases, yet its role in pregnancy related disorders has not been fully elucidated. On the basis of current research, we summarized the changes in important indicators regarding the iron status and ferroptosis in the view of preeclampsia and gestational diabetes mellitus. It may not only provide a new perspective in understanding the pathogenesis of pregnancy-related diseases but also may reveal novel targets for therapeutic strategies. 

### 5.1. In the View of Preeclampsia

Preeclampsia (PE) is a hypertensive disorder during pregnancy that causes severe maternal and fetal morbidity. It is characterized by impaired trophoblast invasion and incomplete spiral artery remodeling, which results in placental ischemia and elevated systemic blood pressure [110]. The etiology of PE remains uncertain, yet the role played by dysregulated iron status has been investigated in recent studies. Levels of serum iron, ferritin, transferrin saturation, and hepcidin were found to be significantly elevated in PE patients compared with healthy pregnant controls [111,112,113]. Furthermore, serum and placental levels of lipid peroxides along with the products of lipid peroxidation such as malondialdehyde (MDA) have shown significant correlations with severe PE [112]. It was speculated that the ischemic placenta may be the primary source of toxic iron that contributes to endothelial cell injury. Elevated hepcidin level could be viewed as a protective mechanism against iron overload and antioxidant supplementation might be beneficial in the scenario of PE. 

At the early onset of pregnancy, placental perfusion is obstructed by trophoblast cell debris and blood clots. At around 10–12 weeks of gestation, maternal blood perfusion is initiated in the intervillous space, resulting in a steep increase in glucose and oxygen [114]. Much like the process of ischemia-reperfusion injury, the rapid reperfusion causes massive oxidative stress and increases the risk of ferroptosis in trophoblasts, underlying the pathogenesis of placental dysfunction [115]. In accordance with this, both placental GSH level and GPX4 activity were reduced in patients with PE comparing to those in healthy controls. Serum selenium, which is indispensable for the function of GPX4 [116], also showed a significant reduction in PE patients [117]. In an in vitro hypoxia-induced PE model, cell viability was profoundly improved by ferroptosis inhibitors. Moreover, it was demonstrated that the upregulation of miR-30b-5p in PE patients played a pivotal role in ferroptosis by downregulating SLC7A11 and FPN, resulting in decreased GSH level and increased labile iron pool [118]. All results above have suggested a strong correlation between ferroptosis and PE, posing the necessity for further inquiry. 

Ferroptosis and oxidative stress associated with PE may be the consequence of dysregulated antioxidant defense system in trophoblasts [119]. As a major suppressor of ferroptosis, GPX4 is essential for the viability and function of human trophoblasts. Mutations or reduced expression of GPX4 resulted in human placental dysfunction and PE [120,121], while the knockout of Gpx4 in mice exhibited embryonal lethality [122]. Genetic or pharmacological suppression of GPX4 resulted in an increased level of ferroptosis. Another newly identified regulator of trophoblast ferroptosis is PLA2G6 [123]. It belongs to the phospholipase A2 family that is ubiquitously expressed in human placental trophoblasts. PLA2G6 can hydrolyze hydroperoxide phosphatidylethanolamines (Hp-PEs) that are implicated with ferroptosis, thus preventing cells from potential damage [124]. Knockout of PLA2G6 in BeWo cells markedly potentiates ferroptosis induced by RSL3. By silencing GPX4 expression in PLA2G6 knockout and wildtype cells, it was demonstrated that the inhibition of GPX4 and PLA2G6 had a synergistic effect. Exposing Pla2g6ko pregnant mice to hypoxia followed by reoxygenation led to the accumulation of Hp-PEs and increased rate of fetal demise compared to Pla2g6WT [123]. Taken together, these data indicate the protective role of GPX4 and PLA2G6 against ferroptosis and placental dysfunction, suggesting the possibility of anti-ferroptotic therapeutic strategies.

### 5.2. In the View of Gestational Diabetes Mellitus

Gestational diabetes mellitus (GDM) is one of the most common complications of pregnancy with rising prevalence. It is characterized by hyperglycemia developed during pregnancy and resolved after birth, which could cause perinatal consequences as well as long-term metabolic disorders in both mother and fetus. The pathogenesis of GDM mainly lies in the increased insulin resistance and the relative deficient pancreatic beta-cell function when encountering metabolic alterations during pregnancy [125].

Apart from maternal age, ethnicity, and family history of type 2 diabetes mellites, iron status has been identified as another risk factor of GDM in recent studies. In several large prospective cohorts, it was demonstrated that women reporting higher dietary heme iron intake levels during early pregnancy experienced increased GDM risk, while non-heme iron intake seemed to be irrelevant [126,127]. On the basis of data from existing clinical trials, whether iron supplementation correlates with GDM occurrence remains controversial. Some researchers suggested iron supplementation did not increase the risk of GDM [128], whereas others implied periconceptional iron supplementation > 30 mg/d or using >60 mg/d of iron during the second trimester were independent risk factors of GDM [129,130], which may in turn increase offspring birth weight [131]. Moreover, indicators of maternal iron status including serum ferritin, soluble TfR, and hepcidin concentrations or placental expression of FPN were also associated with the risk of GDM [132,133,134,135,136]. The results above indicated that pregnant women with dysregulated iron status were more vulnerable to developing GDM (Table 2), yet the underlying mechanism was unknown. One hypothesis could be based on the observations that iron overload led to oxidative stress and that pancreatic beta-cells were susceptible to oxidative stress [137]. Although elevated serum and urinary levels of oxidative DNA damage products were found in GDM patients [138], direct evidence of impaired pancreatic beta-cell functions needs to be further elucidated.

Hyperglycemia conditions in GDM may result in trophoblast abnormalities. BeWo cells exposed to hyperglycemic conditions exhibited differential expression of cellular iron transporters [139], which could account for the dysregulated iron homeostasis in GDM patients. Furthermore, it was previously proven that high glucose levels could suppress the viability and proliferation of human trophoblast HTR-8/SVneo cells [140]. In a recent study by Han et al., high glucose concentration induced ferroptosis in trophoblasts, which was dependent on the upregulation of Sirtuin 3 (SIRT3), a key regulator of mitochondria oxidative stress. Upregulation of SIRT3 induced the autophagy-dependent ferroptosis by promoting the AMPK-mTOR pathway as well as decreasing cellular GPX4 level [141]. Investigating the role of ferroptosis in trophoblasts should be of interest since it could bring new insights to understanding the pathogenesis of GDM.

## 6. Conclusions and Future Perspectives

Iron is an essential element for numerous biological processes. The absorption, distribution, and metabolism of iron are tightly regulated on both systemic and cellular levels. During normal pregnancy, iron plays a pivotal role in supporting maternal requirements, placental functions, and fetal developments, while dysregulated iron status is associated with the occurrences of several pathological conditions. In this review, we mainly summarized the transport and regulation of iron in the placenta and revealed the role of iron metabolism and ferroptosis in the view of preeclampsia and gestational diabetes mellitus on the basis of current knowledge.

Despite decades of hard effort, our understanding of placental iron trafficking is still very limited. Although a large number of iron transporters were found to be highly expressed in human and rodent placentas, very few of them were localized to syncytiotrophoblasts. Further research using placental-specific knockout mice should help to elucidate their significance in placental iron transport. Furthermore, given that most current studies were focused on transferrin-bound iron transport, there is a huge blank regarding the uptake and metabolism of other iron sources in the placenta. In addition, it is worthwhile to investigate how the placenta senses and integrates the systemic and local signals that regulate iron trafficking and whether other iron regulators exist in parallel with hepcidin. Lastly, as a newly discovered iron-related programmed cell death pathway, ferroptosis has been a research topic of growing interest. Exploring how ferroptosis participates in the pathogenesis of pregnancy related diseases would benefit the prevention and treatment of such conditions.

## Figures and Tables

**Figure 1 ijms-23-09395-f001:**
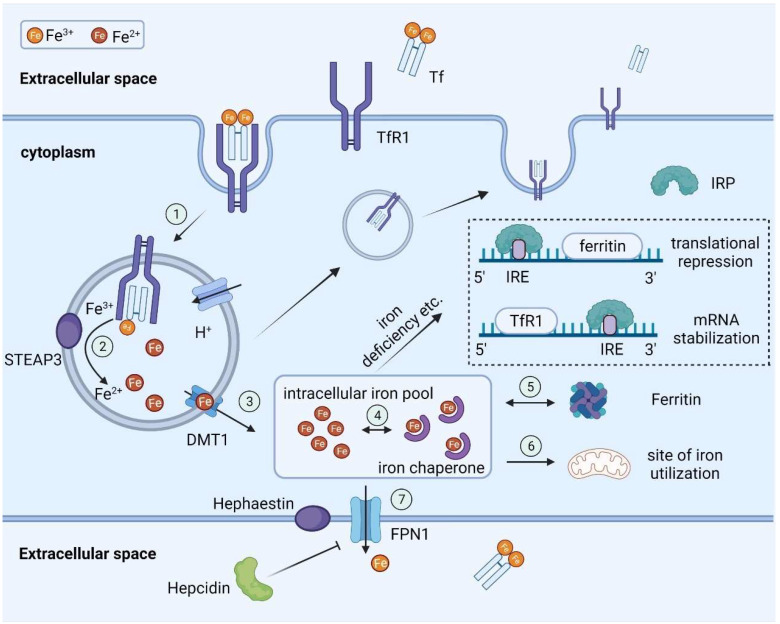
Cellular iron transport and regulation under physiological conditions. Iron transport: Plasma diferric Tf binds with TfR1 expressed on the membrane of cells. The Tf–TfR1 complex is internalized by receptor-mediated endocytosis (**1**). Within the highly acidified endosome, iron is released from Tf after the reduction of ferric iron takes place with the help of ferrireductase STEAP3 (**2**). Iron crosses the endosomal membrane into the cytoplasm via DMT1 (**3**). Iron chaperones exist in the cytoplasm to move iron within the cell (**4**). Intracellular iron can either be sequestered with ferritin (**5**) or transported directly to the sites of need (**6**). Upon mobilization, iron can efficiently detach from ferritin and be exported by FPN (**7**). Iron regulation: Hepcidin blocks cellular iron egress by binding with FPN and leads to its degradation. Intracellular iron level can be regulated by the IRP-IRE system. When cellular iron concentration is low, IRPs would bind to the 5′ UTR of ferritin mRNA and the 3′ UTR of TfR1 mRNA, which blocks the translation of the former and protects the latter from degradation. *Abbreviations*: Tf, transferrin; TfR1, transferrin receptor 1; STEAP3, six-transmembrane epithelial antigen of the prostate 3; DMT1, divalent metal transporter 1; FPN, ferroportin; IRP, iron regulatory protein; IRE, iron-responsive element.

**Table 1 ijms-23-09395-t001:** Function and localization of iron transporters in placenta.

Protein (Gene)	Function	Localization in Human Placenta	Disruption and Phenotype
TfR1 (*TFRC*)	TBI and ferritin uptake	On the apical membrane of STB [39]	*Trfc*^−/−^ was embryonic lethal at E12.5; *Trfc*^+/−^ mice showed severe anemia [43]
DMT1 (*DMT1; SLC11A2*)	Intracellular iron trafficking	Prominently near maternal side; rarely on fetal side; scatter staining in stroma [44]	*Slc11a2*^−/−^ mice were pale at birth; not iron deficient in all tissues comparing to wildtype littermates; *Slc11a2*^+/−^ mice were viable without visible abnormalities [45]
ZIP14 (*SLC39A14*)	NTBI uptake and intracellular iron trafficking	Not available	*SLC39A14*^−/−^ mice were viable with growth retardation; iron relating parameters not reported [46]
ZIP8 (*SLC39A8*)	NTBI uptake and intracellular iron trafficking	Not available	*SLC39A8*^neo/neo^ newborns were pale and growth was stunted with diminished iron uptake [47]
Hephaestin (*HEPH*)	Ferroxidases	Not available	Mice with global or intestine-specific knockout of *Heph* were viable with microcytic anemia due to reduced intestinal iron absorption [48,49]
Ceruloplasmin (*CP*)	Ferroxidases	Intervillous space [50]	Although normal at birth, *Cp*^−/−^ mice showed progressive accumulation of iron in liver and spleen, but not in macrophages [51,52]
Zyklopen (*ZP*, *HEPHL1*)	Ferroxidases	Cytoplasm of STB [53]; maternal decidua [54]	*Zp*^−/−^ showed increased placenta size with no change in fetal iron transfer [54]
SCARA5 (*SCARA5*)	Ferritin uptake	Not available	Not available
FLVCR1 (*FLVCR1*)	Heme uptake	Not available	*FLVCR1*^−/−^ was embryonic lethal at E7.5 due to impaired erythropoiesis [55]
BCRP (*ABCG2*)	Heme uptake	On the apical membrane of STB [56]	*ABCG2*^−/−^ mice were viable; iron relating parameters not reported [57]
LRP1 (*LRP1*)	Heme uptake	Not available	*LRP1*^−/−^ was embryonic lethal at E12.5 [58]
HO-1 (*HO1*)	Heme iron metabolism	In STB and cytotrophoblasts [59]	*HO1*^−/−^ decreased embryo viability; *HO1*^+/−^ led to placental dysfunction; iron relating parameters not reported [60]
FPN (*FPN*, *SLC40A1*)	Iron export	On the basolateral membrane of STB	*SLC40A1*^−/−^ causes embryo lethality before E7.5; *Meox2-Cre; Fpn^flox/flox^* mice, in which FPN was only expressed in placenta, were viable with anemia and cellular iron accumulation [10]

*Abbreviations*: TfR1, transferrin receptor 1; TBI, transferrin-bound iron; STB, syncytiotrophoblasts; E, embryonic day; DMT1, divalent metal transporter 1; ZIP, Zrt/Irt-like protein; NTBI, non-transferrin-bound iron; SCARA5, scavenger receptor class A member 5; FLVCR1, feline leukemia virus subgroup C receptor 1; BCRP, breast cancer resistance protein; LRP1, LDL-receptor-related protein 1; HO, heme oxygenase; FPN, ferroportin.

**Table 2 ijms-23-09395-t002:** Prospective studies reporting iron-related risk factors of GDM.

Factors	Study	Research Design	Comparison Groups	Adjusted RR/OR (95% CI)
Dietary heme iron intake	Qiu et al. (2011) [126]	Prospective cohort; 3158 pregnant women	Heme iron intake levels (≥1.52 vs. <0.48 mg per day)	3.31 (1.02–10.72)
	Bowers et al. (2011) [127]	Prospective study; 13,475 pregnant women	Median heme iron intake levels (1.60 vs. 0.66 mg per day)	1.58 (1.21–2.08)
Dietary non-heme iron intake	Qiu et al. (2011) [126]	Prospective cohort; 3158 pregnant women	Non-heme iron intake levels (≥12.98 vs. <0.10 mg per day)	0.61 (0.31–1.18)
	Bowers et al. (2011) [127]	Prospective study; 13,475 pregnant women	Median heme iron intake levels (45.33 vs. 7.58 mg per day)	0.97 (0.78–1.20)
Iron supple-mentation	Bowers et al. (2011) [127]	Prospective study; 13,475 pregnant women	Median Iron supplementation levels (60.00 vs. 0 mg per day)	1.04 (0.84–1.28)
	Chan et al. (2009) [128]	RCT; 1164 pregnant women with Hb level between 8–14 g/dl	60 mg daily iron supplementation vs. placebo group	1.04 (0.70–1.53)
	Zhang et al. (2021) [129]	Prospective cohort; 2117 pregnant women	>60 mg daily iron supplementation during the second trimester vs. non-users	1.43 (1.06, 1.92)
	Zhang et al. (2021) [130]	Prospective cohort; 5101 pregnant women	>30 mg daily iron supplementation for more than 3 months vs. non-users	1.53 (1.21–1.93)
Serum ferritin	Rawal et al. (2017) [134]	Prospective case–control study; 107 women with GDM and 214 controls	Highest vs. lowest quartile of serum ferritin level	2.43 (1.12–5.28)
	Bowers et al. (2016) [135]	Prospective case–control study; 350 women with GDM and 349 controls	Highest vs. lowest quartile of serum ferritin level	2.22 (1.23–4.01)
	Khambalia et al. (2016) [136]	Prospective cohort study; 4420 pregnant women	Serum ferritin level <12 μg/L vs. normal	0.43 (0.23–0.78)
Serum sTfR	Rawal et al. [134]	Prospective case–control study; 107 women with GDM and 214 controls	Highest vs. lowest quartile of serum sTfR level	1.00 (0.45–2.20)
	Bowers et al. [135]	Prospective case–control study; 350 women with GDM and 349 controls	Highest vs. lowest quartile of serum sTfR level	1.48 (0.82–2.70)
	Khambalia et al. (2016) [136]	Prospective cohort study; 4420 pregnant women	Serum sTfR level >21 nmol/L vs. normal	1.25 (0.82–1.92)
Serum hepcidin	Rawal et al. (2017) [134]	Prospective case–control study; 107 women with GDM and 214 controls	Highest vs. lowest quartile of serum hepcidin level	2.61 (1.07–6.36)

*Abbreviations*: GDM, gestational diabetes mellitus; sTfR, soluble transferrin receptor.

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
