# Peer review of "Iron Metabolism and Ferroptosis in Physiological and Pathological Pregnancy"

_ijms, 2022, doi:10.3390/ijms23169395_

Round 1
Reviewer 1 Report
Congratulations on your very interesting paper. Of course it raises more questions than gives an answers but the concept of the influence of iron metabolism on possible maternal and fetal/neonatal health condition and outcome is really intriguing. However the level of recent scientific evidence is low and huge studies should be performed in the future.
Some remarks about article:
As your article is review article you have to present methodology of the literature analysis: time frame, search data bases, key words, inclusion and exclusion criteria, language etc.
Author Response
Some remarks about article: As your article is review article you have to present methodology of the literature analysis: time frame, search data bases, key words, inclusion and exclusion criteria, language etc.
Reply: We greatly appreciate the reviewer’s positive comments on our review. We have added methodology description in our manuscript from line 53 to 60: A search of PubMed database was performed to identify relevant literature written in English focusing on iron metabolism and pregnancy. The search term “iron metabolism” was combined with a range of terms related to the focus of the review, such as pregnancy, placenta, placental trafficking, homeostasis, systemic regulation, ferroptosis, preeclampsia, and gestational diabetes mellitus, etc. The search included studies from 1975 to 2022. After title and abstract screening for study type, case reports, case series, and animal studies with non-rodents were excluded. The reference list was further checked by full-text reading and 141 articles were included in the final review.
Reviewer 2 Report
A carefully prepared narrative review highlighting the latest scientific achievements in the field of iron metabolism in pregnancy.
I have two minor comments:
1. I ask the authors to consider the discussion of the relationship of iron deficiency with disorders of the nervous system of newborns and the possible association of NTBI with CNS damage.
2. Please discuss the following manuscripts:
https://doi.org/10.3390/ijms23158294
DOI: 10.3109/14767058.2012.735999
DOI: 10.3390/nu14132624.
DOI: 10.3390/nu13072480.
Author Response
1. I ask the authors to consider the discussion of the relationship of iron deficiency with disorders of the nervous system of newborns and the possible association of NTBI with CNS damage.
Reply: We greatly appreciate the reviewer’s positive comments on our review. We have added a paragraph (from line 165 to 177) discussing the effects of gestational iron deficiency on the fetus, which mainly focused on the fetal nervous system: Moderate to severe gestational iron deficiency could lead to both immediate and long-term consequences to the offspring. Given the important role played by iron in neurodevelopment, brain structure alterations and cognitive impairments are the most common short-term risks. Evidence from animal models and human observational studies have demonstrated that the hippocampus and the process of myelination are particularly vulnerable to iron deficiency. The imbalance between high energy demand in these rapidly growing structure and impaired cellular energy metabolism in the lack of iron could be the possible explanation. Moreover, gestational iron deficiency may also be associated with long-term mental health disorders such as depression, anxiety, autism and schizophrenia in offspring. This could be accounted by decreased monoamine production with impaired emotion and memory process due to iron deficiency. Iron supplementation during gestational and postnatal period may be beneficial to neurodevelopment.
As for the association of NTBI with CNS damage, the majority of current research focused on how NTBI led to neurodegenerative disease. There is not much literature discussing the relationship between maternal NTBI and fetal brain damage given that whether NTBI can be transported by the placenta is still controversial. So, we could not elaborate on this topic and just added “Investigation on this topic would be of great importance to patients in iron overload conditions since NTBI may be untaken by fetal neurons, resulting in generation of ROS and neonatal encephalopathy.” on line 229. We would be grateful if you could hint us on how we should discuss this topic.
2.Please discuss the following manuscripts:
https://doi.org/10.3390/ijms23158294
DOI: 10.3109/14767058.2012.735999
DOI: 10.3390/nu14132624.
DOI: 10.3390/nu13072480.
Reply: Thank you for the suggestions. All the recommended references have been included in the proper section of our revised manuscript (Ref 26,32, 37,131).